# Intensive Care Clinicians’ Perspectives on Ethical Challenges Raised by Rapid Genomic Testing in Critically Ill Infants

**DOI:** 10.3390/children10060970

**Published:** 2023-05-30

**Authors:** Sachini Poogoda, Fiona Lynch, Zornitza Stark, Dominic Wilkinson, Julian Savulescu, Danya Vears, Christopher Gyngell

**Affiliations:** 1Department of Paediatrics, University of Melbourne, Melbourne, VIC 3010, Australia; 2Murdoch Children’s Research Institute, Royal Children’s Hospital, Melbourne, VIC 3052, Australia; 3Australian Genomics, Melbourne, VIC 3052, Australia; 4Faculty of Philosophy, Oxford Uehiro Centre for Practical Ethics, University of Oxford, Oxford OX1 1PT, UK; 5Centre for Biomedical Ethics, Yong Loo Lin School of Medicine, National University of Singapore, Singapore 119077, Singapore; 6Centre for Biomedical Ethics and Law, KU Leuven, 3000 Leuven, Belgium

**Keywords:** rapid genomic testing, paediatric intensive care, intensivist, global standards, critically ill infants

## Abstract

Rapid genomic testing (rGT) enables genomic information to be available in a matter of hours, allowing it to be used in time-critical settings, such as intensive care units. Although rGT has been shown to improve diagnostic rates in a cost-effective manner, it raises ethical questions around a range of different areas, including obtaining consent and clinical decision-making. While some research has examined the perspectives of parents and genetics health professionals, the attitudes of intensive care clinicians remain under-explored. To address this gap, we administered an online survey to English-speaking neonatal/paediatric intensivists in Europe, Australasia and North America. We posed two ethical scenarios: one relating to obtaining consent from the parents and the second assessing decision-making regarding the provision of life-sustaining treatments. Descriptive statistics were used to analyse the data. We received 40 responses from 12 countries. About 50–75% of intensivists felt that explicit parental consent was necessary for rGT. About 68–95% felt that a diagnosis from rGT should affect the provision of life-sustaining care. Results were mediated by intensivists’ level of experience. Our findings show divergent attitudes toward ethical issues generated by rGT among intensivists and suggest the need for guidance regarding ethical decision-making for rGT.

## 1. Introduction

Genetic conditions and congenital abnormalities are a leading cause of infant mortality in developed nations [1,2,3,4]. In recent years, genome and exome sequencing technology (collectively referred to as genomic sequencing) has revolutionised care for critically ill children with rare genetic diseases [1,5,6]. As genomic sequencing technology has evolved, the time required to sequence and analyse a genome has fallen dramatically [2]. Where previously the return of genomic sequencing results could take six months [5], Rapid Genomic Testing (rGT) has reduced this to weeks or days [2] and more recently to hours [7]. This reduced turnaround time has meant genomic sequencing can be applied to time-critical settings, such as for patients in neonatal and paediatric intensive care units (NICU and PICU). In the NICU, rGT achieves a diagnostic yield between 30% and 52% [8,9,10] and is a cost-effective and practical option for testing [2,5,10,11,12,13].

There have been several studies exploring parental perspectives of rGT testing for infants where, in general, parents show low decision regret and felt that they were more informed about their child’s condition after testing [14,15,16,17,18,19,20].

The clear benefits associated with rGT have led to calls for it to be a first-line test for children in intensive care units around the world [21,22]. However, there remain ongoing questions about how such a complex test could be implemented in this setting. Some have expressed concern about the lack of availability of specialist genetics health professionals (GHPs) in a 24/7 emergency setting [23]. This means that, in the future, paediatric and neonatal intensivists are likely to be the ones tasked with administering rGT. Currently, in many major children’s hospitals, an on-call clinical geneticist is likely accessible for consultation via telehealth or telephone. However, as rGT becomes more widely offered, paediatric and neonatal intensive care clinicians (whom we refer to collectively as intensivists) will be the clinicians obtaining parental consent for rGT and make treatment decisions based on rGT outcomes without immediate support from GHPs. Furthermore, exact processes and professional responsibilities vary across hospitals and jurisdictions. Examining the views of intensivists about these ethical issues is crucial to gain insights into how to best support them as rGT becomes implemented more widely.

Some previous work has explored the perspectives and experiences of intensivists with rGT. In 2019, [24] explored the attitudes of 21 neonatal intensivists with some experience using genomic technologies in Kansas City, Missouri, towards the use of rGT in the NICU. Clinicians expressed concern about how to interpret rGT results and how and why genomic results could be clinically useful. They discussed the potential harms of genomic testing including the impact on future insurance policies and receiving undesired information and questioned if parental consent was always necessary [24]. Previous work at the Murdoch Children’s Research Institute (MCRI) involved focus groups with health professionals involved in the delivery of rGT [25]. These highlighted that due to the rapidity of administering rGT and receiving results, consent and withdrawal of life-sustaining treatment were ethically challenging areas for clinicians. rGT increased the number of cases where life and death decisions were being made with little time to reflect on the underlying issues.

Globally, there are several professional bodies that provide guidelines for genetic testing, such as the American College of Medical Genetics and Genomics (AMCG), the Human Genetics Society of Australasia (HGSA), the Canadian College of Medical Geneticists (CCMG) and the European Society of Human Genetics (ESHG). Each provides guidance for the use of genomic sequencing technologies in their region and establishes ethical standards for practitioners when ordering predictive and diagnostic genetic testing [26,27,28,29]. However, these bodies do not yet provide guidance for rGT and if recommendations for obtaining consent and storing DNA should differ in the rapid space [30].

Although previous research has identified ethical challenges for implementing rGT, how these challenges should be managed in practice has not been explored. In this study, we asked intensivists about their attitudes toward ethically challenging situations that could arise as a result of rGT. Specifically, we explored (1) whether it is ever acceptable to conduct rGT without explicit parental consent; (2) what types of results from rGT influence decisions about lifesaving treatment and (3) if these attitudes differed between intensivists practising in different countries and across years of experience.

## 2. Materials and Methods

### 2.1. Study Design

The survey (see Appendix A) was developed by SP, with assistance from CG, DV and FL, based on the findings from focus groups conducted with health professionals using rGT in the acute care setting [25]. Case scenarios (Box 1) were developed for the survey, with questions focusing on topics of consent and administering/withholding treatment in the context of rGT. Questions about withholding treatment centred around three diagnoses (STRA-6-related disorders, Alagille syndrome and Kabuki syndrome) ranging in spectrum from mild to severe physical and intellectual impact. Feedback was sought from subject matter experts (ZS and JS) on the wording and clinical details of the conditions in the survey. These changes were implemented and reviewed before the survey was piloted with an intensivist and an intensivist/ethicist (DW).

Box 1The ethical scenarios posed in the survey provided context for questions that focussed (A) on topics of consent and (B) administering/withholding treatment in the context of rGT.
*Scenario A:*

*Alex was born prematurely at 36 weeks’ gestation with multiple dysmorphic features and complex*

*congenital anomalies, which will require surgery. The intensive care team decide rapid genomic testing is most likely way to identify a diagnosis and avoid any unnecessary invasive procedures.*

*Scenario B:*

*Sam was born at 32 weeks with multiple dysmorphic features, and a complex heart condition for which surgery is indicated Surgery has a 50%chance of successfully treating the heart problem, though the overall prognosis is unclear. Rapid genomic sequencing is ordered with the hope of learning more about the prognosis. The test identifies two mutations in the STRA6 gene which cause a recessive syndromic disorder associated with alveolar capillary dysplasia, diaphragmatic eventration, microphthalmia and profound intellectual disability. This means that even if Sam survives the cardiac surgery, there is a 95% chance they will die in the first year of life, likely secondary to pulmonary issues.*


### 2.2. Recruitment

The survey was distributed to intensivists through professional networks, mailing lists and social media. The survey was forwarded to all Heads of NICU Departments in Victoria, Australia and their teams; posted in the World Federation of Paediatric Intensive and Critical Care Society newsletter and forwarded to the British Association of Perinatal Medicine mailing list. These lists targeted English-speaking paediatric and neonatal intensivists primarily from Europe, Australasia and North America. A post was also made on Twitter through the Murdoch Children’s Research Institute Biomedical Ethics Research Group account and retweeted by members of the research team with large followings, including some intensivists.

### 2.3. Data Collection and Analysis

Data were collected via the online survey tool REDCap [31], and analysis was completed using R 2022.07.0 [32]. Responses from clinicians not working in intensive care or responses with only demographic information were discarded. Frequency and percentages were calculated for categorical data (demographic data and yes/no responses to ethical questions). Chi-squared analyses were undertaken for comparisons between demographic data and responses to multiple-choice questions. Answers to open-ended questions were analysed by SP using an inductive content analysis [33] and co-coded by DV. Free-text responses are reported below to help explain intensivists’ reasoning for their answers; an illustrative quote for each question is included.

## 3. Results

### 3.1. Demographics

There were 40 responses in total. Thirty-nine respondents were neonatal intensivists or trainees; of these, two were also paediatric intensivists/trainees, one was also a medical geneticist and one was also an ethicist. The other respondent was a paediatric pulmonologist involved in intensive care. Responses were received from 12 countries; most (72.5%) were from Australia, the United Kingdom (UK) and the United States (US) (Table 1).

Respondents had a wide range of experience in the NICU/PICU, with 27% of respondents having 20 or more years of experience (Table 1). Respondents also had a range of experience ordering genetic tests; 22.5% had ordered more than 20 standard genetic tests but 65% had ordered less than four rapid genomic tests (Table 1).

### 3.2. Obtaining Parental Consent in the Context of Rapid Genomic Testing—Responses to Scenario A

Half (*n* = 20) of respondents thought that consent should be obtained from parents by a GHP even if this meant waiting an extra day and potentially placing the patient at greater risk. The remainder thought that consent should be obtained by whichever non-GHP was available. In the free-text comments, intensivists affirmed that clinicians obtaining consent needed to have training and sufficient understanding of the complexities of genetic testing, and that the consenting clinician should be able to adequately address parent questions and provide information. They also suggested it is important to balance the need for a specialist obtaining consent, with the impact a delay may have on patients. 


*“The decision may also be influenced by how rapidly he [Alex] needed life sustaining surgery and how soon a geneticist could get there.”—P22, neonatal intensivist, Canada, 0–4 years of experience.*


Overall, respondents were divided on when explicit consent should be obtained from parents (Table 2).

Several respondents suggested that intensivists and other healthcare professionals working in the NICU should receive some training in pre-test counselling from GHPs.


*“Healthcare professionals taking consent for genomic testing should have had training from genetic colleagues and only take consent if confident to do so, otherwise consent taking should be supported by a genetics health professional.”—P40, neonatal intensivist, United Kingdom, 15–19 years of experience.*


Sixty-five percent of respondents (*n* = 26) thought that the intensive care team should wait for one of the parents to be available to provide consent before the treating clinician ordered rGT for Baby Alex, and a quarter (*n* = 10) thought that the treating clinician should be able to order rGT without parental consent. Ten percent (*n* = 4) thought that both parents should be available to provide consent before testing proceeded, even though the delay may have meant the infant’s condition deteriorated.

In the free-text comments, intensivists discussed that a DNA/blood sample should be taken for storage from the infant and testing arranged while awaiting consent from the parents. They discussed that trio testing would be preferable regardless, so it would be best if both parents were able to provide consent. They suggested clinicians should consider how time delays may affect the child’s clinical picture. Respondents also raised the idea that intensivists should be allowed to perform lifesaving emergency interventions, including genetic testing, until the parents could be consulted, as rGT was no different from any other test.


*“Having a child admitted to a NICU infers consent for ‘usual treatment’. Discovery of an underlying genetic cause for a condition is ‘usual treatment’. It is only that the technology being used is different.”—P35, neonatal intensivist, Australia, 20+ years of experience.*


In the event that one parent refused to give consent for rGT for Baby Alex, 57% (*n* = 23) of respondents thought that rGT should not proceed. Respondents discussed that testing should go ahead when in the best interests of the infant, and that family relationships also needed to be considered, as well as trust between the clinicians and the family. They highlighted the potential long-term impact of proceeding without both parents’ consent.


*“Clear disagreement may result in further harms to this family.”—P3, neonatal intensivist, Armenia, 5–9 years of experience.*


In the scenario where both parents were overwhelmed and had said yes to testing, but the clinician is concerned that they are unable to give informed consent, 55% (*n* = 22) agreed that rGT should proceed. Intensivists discussed that the clinical team needed to support parents to make decisions, the clinician’s responsibility was to advocate for the child and the consent in the NICU setting is rarely informed.


*“It is common for families in our care to be overwhelmed and distressed. We must trust that they are making the best decision they can at the time.”—P8, neonatal intensivist, Australia, 10–14 years of experience.*


In the scenario in which Baby Alex’s parents do not speak English, 72% (*n* = 29) felt that rGT should not proceed until an interpreter was available to speak with the family. Respondents discussed that phone or internet interpreter services should be used when possible, and that Alex’s parents need to understand the testing for it to proceed.


*“If consent is being sought–testing cannot go ahead without being able to verify that parents understand.”—P2, neonatologist, United Kingdom, 15–19 years of experience.*


### 3.3. Withdrawal of Life-Sustaining Treatment in the Context of Rapid Genomic Testing—Responses to Scenario B

In the scenario where rGT reveals that Sam has a 95% chance of death in his first year of life, 95% (*n* = 38) of intensivists thought that surgery with a 50% chance of being saved should not proceed. In free-text comments, intensivists discussed considering the overall benefits for Sam’s quality of life, the collaborative discussion with his parents was required and the consideration of Sam’s clinical outcomes was needed to make a decision. 


*“It may be appropriate for a relatively simple procedure to be done but the focus should be on this child’s quality of life. Care should be directed towards making his short life as happy and distress free as possible.”–P39, neonatologist, United Kingdom, 20+ years of experience.*


After Baby Sam received the diagnosis of a STRA6-related disorder, 95% (*n* = 38) of respondents felt that Sam’s parents should be able to refuse cardiac surgery for their child. All but one intensivist who felt surgery should not proceed thought that Sam’s parents should have the right to refuse surgery.

In the scenario where Sam was instead diagnosed with Alagille syndrome (which means he would require liver transplantation in childhood and lifelong immunosuppression but have a normal intellectual function), 85% (*n* = 34) of respondents thought that cardiac surgery should proceed. Sixty-eight percent (*n* = 27) of respondents thought that the parents should be able to refuse cardiac treatment when Sam’s diagnosis was Alagille syndrome. There was an association between the increased experience level of respondents and allowing the parents to refuse treatment (X^2^ (4, *n* = 40) = 14.58, *p* = 0.012) (Table 3). Respondents with less than nine years of experience were evenly divided between whether the parents should be allowed to refuse treatment or not. However, all respondents (*n* = 7) with 10–14 years of experience and 10 of 11 with more than 20 years of experience thought that the parents should be able to refuse treatment.

Intensivists discussed that, in this scenario, parents should be involved in the decision-making and that a wider review with an ethics board may be needed. They also said it is important to consider Sam’s quality of life and suffering and that a decision should be made for the best outcome of Sam’s clinical picture.


*“These are 2 serious conditions—it depends on exactly what the complex heart disease is and the morbidity associated with that. I think it’s a finely balanced decision whether to operate or not—and his parents should be involved in that decision.”—P39, neonatologist, United Kingdom, 20+ years of experience.*


In the scenario where Sam was instead diagnosed with Kabuki syndrome, meaning that he will not die in infancy but is likely to have moderate to severe intellectual disability, 60% (*n* = 24) of respondents felt that cardiac surgery should proceed and 80% felt that Sam’s parents should be allowed to refuse treatment. There was an association between the clinician’s continent of practice and response to this question (X^2^ (2, *n* = 40) = 6.84, *p* = 0.033) (Table 3). Australasian and North American intensivists were more likely than European intensivists to say Sam’s parents should be able to refuse treatment. Respondents discussed, respecting parental wishes, the need to account for the variable presentation of Kabuki syndrome and again raised that referring to a clinical ethics response group may be necessary.


*“Will definitely need help from geneticist and ethical committee.”—P14, neonatologist, Belgium, 5–9 years of experience.*


An overall trend was that as the number of intensivists who supported cardiac surgery being performed increased, they were less likely to be in favour of the family being able to refuse cardiac surgery (Figure 1).

## 4. Discussion

This study is the first to provide insight into the global perspectives of intensivists towards ethically ambiguous scenarios involving rGT for critically ill infants. Our major finding is that intensivists were divided on when informed consent is explicitly required during rGT. Overall, our findings indicate that if rGT is implemented worldwide, there will potentially be significant differences in how intensivists approach parental consent and variability in how parents are treated between different hospitals depending on the continent and the intensivist’s level of experience. This underlines the need for formal guidance to guide ethical decision-making when using rGT for critically ill children.

### 4.1. Intensivists Are Divided on Whether Rapid Genomic Testing Requires Specific Consent

A prevalent belief is that genomic sequencing for minors ought to be carried out solely with the approval of the patient’s parents, and any guidance provided by a health professional should aim to defer testing to adulthood when possible [34]. However, the use of rGT in critically ill children challenges this paradigm [35]. Unlike other contexts where genomic sequencing is performed, even small delays due to consent for rGT can have long-term health implications for children. Doctors in intensive care units are focussed on improving the health of infants through all the tools at their disposal. For diagnostic tests, even invasive ones such as lumbar puncture, consent is often presumed. Some have argued that because of the high clinical utility of rGT in this setting, parental consent for rGT should also be presumed [35,36].

The findings in our study show that intensivists are split on the question of whether explicit parental consent is necessary for rGT. Around a quarter thought that it was acceptable to perform rGT in a range of circumstances, where explicit parental consent was absent. Half thought that explicit parental consent was necessary across all circumstances presented. Another quarter thought that parental consent was necessary in some, but not all the scenarios presented. This finding aligns with Knapp et al.’s (2019) observation of uncertainty among intensivists regarding obtaining detailed consent for genomic testing in NICU settings. 

If rGT was widely implemented now, it is likely that the degree to which explicit parental consent was viewed as necessary would vary. Some intensivists would likely consider parental consent for rGT to be presumed as part of consent for “usual treatment” in NICUs and PICUs. Others would likely only perform rGT when there is explicit parental consent, facilitated through a GHP. Broader engagement work and ethical analyses may be necessary for building consensus regarding the role of parental consent in rGT.

### 4.2. Impact of Genetic Diagnosis on Life-Sustaining Care

The use of rGT in the NICU presents complex dilemmas concerning the impact of specific genetic diagnoses on parental and clinician decisions to withhold or limit treatment [35]. A traditional example of treatment limitation in children with a genetic condition is Down syndrome. This congenital disorder can be quickly identified through clinical examination and validated via fluorescent in situ hybridization within a day. In the past, a Down syndrome diagnosis often led to the withholding of potentially lifesaving cardiac surgery [37]. However, opting not to perform surgery on infants with Down syndrome, while offering it to patients without Down syndrome, has been criticized as discriminatory [38]. As a result, the current standard practice is to provide equal opportunities for cardiac repair.

The widespread implementation of rGT will lead to many more genetic conditions being diagnosed early. An ethical issue that might emerge from the implementation of rGT is uncertainty regarding whether a diagnosis of a genetic disease should affect offers of potentially life-extending treatments, such as cardiac repair, as well as the position of parents to refuse potentially lifesaving surgeries [35,39].

Kabuki syndrome, like Down syndrome, is a genetic condition that is associated with congenital heart defects and varying degrees of developmental delay and intellectual impairment. We found that 40% of intensivists thought that cardiac repair should not be offered in the cases where rGT has led to a diagnosis of Kabuki syndrome, and 80% thought that parents should be allowed to refuse cardiac surgery in this case. If rGT was widely implemented and led to more infants with lifelong developmental disabilities not having life-extending surgery, this could be seen as ethically mirroring the already criticised practice of not offering cardiac repair to patients with Down syndrome. 

When a genetic condition was associated with death in the first year of life, only one of the intensivists surveyed thought that potentially lifesaving surgery should be performed. This suggests that when genetic conditions are severely life-limiting, there is relative consensus about the ethical acceptability of not offering life-extending treatment. 

In the scenario where the condition is associated with severe health impacts but is treatable and is not associated with developmental delay (Alagille syndrome), a clear majority of intensivists (85%) thought that the diagnosis should not impact life-extending surgery. The differences in attitudes of intensivists toward the implications of Kabuki and Alagille diagnoses could be explained by the fact that Alagille syndrome is treatable and not associated with developmental delay.

Char, Lee [17] found that intensivists envisioned the largest potential benefit of rGT in the NICU to be earlier guidance involving the withdrawal of life-sustaining treatment. Our study adds depth to this finding. If rGT identifies a severe, life-limiting condition, there is broad consensus among intensivists about how this diagnosis should influence end-of-life treatment decisions. However, this finding also suggests that if rGT leads to the diagnosis of a less severe disease, there may be disagreements about how such a diagnosis affects the provision of life-sustaining care. 

More research is needed to explore this point. Wider agreement and consensus on which features of a genetic diagnosis are relevant to withholding/withdrawal of treatment decisions would help to support the appropriate incorporation of genomic testing into these critical decisions.

### 4.3. Impact of Demographic Differences on Parental Discretion

Intensivists with more years of experience were more likely to allow parental discretion in the decision to proceed with rGT or surgery. Though not representative of universal guidelines, current National Health Service guidelines in the UK state that decisions about care and treatment must be made in the child’s best interests in partnership with parents [40]. These guidelines suggest that more experienced intensivists may favour parental discretion, as more experience leads to greater skill at navigating disagreements and mediation, leading to increasingly patient-centred practice.

European intensivists were also more likely to be in favour of surgery, and Australasian and North American respondents were more likely to favour parental discretion in the case of a diagnosis that included many body systems and intellectual disability. Though the sample sizes of each of these groups were small, this could be indicative of differing schools of thought between continents. 

### 4.4. Study Limitations

This study was limited by the number of responses from each demographic group and the number of responses overall. More responses would be useful to further understand the differences between respondents from different countries and intensivists with more/less years of experience. Other surveys of intensivists have collected views from 30 to over 1000 potential respondents [41,42,43]. Increased response number would allow dividing these responses into specific demographic groups, such as countries, which may show differences in values where laws regarding parental involvement in consent and genetic testing differ and allow more direct conclusions to be made. 

The study design also forced intensivists to make choices between discrete categories, which may not reflect the nuance of real-world situations. Different intensivists may have made different assumptions about the background to each case. 

This survey did not explore the intricacies of intensivists’ understandings of the consent process as outlined in American College of Medical Genetics (ACMG) guidelines (2013). These provide guidance on what specific topics to cover during the consent process, including the possibility of incidental findings and what kinds of results will/will not be returned [44]. While this study indicated that most considered facilitating understanding an integral part of informed consent, the survey did not address what other gaps may exist in the consent process if undertaken by an intensivist. This study did not include other challenging aspects of delivering rGT, such as how to counsel families around uncertain results. Future qualitative research may be useful to understand these nuances.

## 5. Conclusions

This novel study into intensivists’ perspectives on using rGT for critically ill infants provided insights into what education and guidelines are required for the integration of rGT into NICU/PICU settings. Intensivists worldwide were divided on when obtaining explicit parental consent for rGT was necessary and treated parental wishes for withdrawal of life-sustaining treatment differently depending on the diagnosis. This suggests that families will be treated differently depending on the country and the treating intensivist’s years of experience if this technology was widely implemented. This supports calls for further research to guide ethical decision-making in this area. For example, the timing of testing must be dictated by the best interests of the child. If a delay in testing would result in significant expected harm to the child, then testing should proceed (both ethically and legally) in the absence of parental consent or even in the face of refusal. If testing can be delayed without significant harm to the child, parental consent should be sought.

Overall, intensivists prioritised obtaining the consent of one or both parents before initiating rGT for infants and considered understanding to be an essential component of informed consent. If rGT was integrated into acute care, it is likely that the informed consent aspect of the ACMG (2013) guidelines would be practised by intensivists. As shared decision-making between families and intensivists was shown to be a particularly important factor in using rGT for critically ill infants, further training for intensivists in facilitating shared decision-making may be necessary to ensure this is upheld. Guidelines are needed for any scenarios where rGT may ethically proceed without specific consent. 

## Figures and Tables

**Figure 1 children-10-00970-f001:**
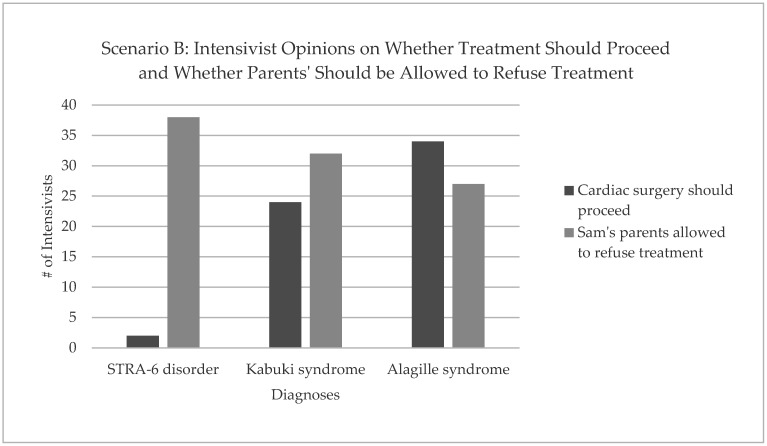
Intensivists’ opinions on whether Sam’s treatment should proceed and whether his parents should be allowed to refuse this treatment.

**Table 1 children-10-00970-t001:** Respondent demographics divided by continent, listing years of experience practising in their speciality and experience ordering standard and rapid genomic tests.

	Australasia	Europe	North America	All
	n (%)	n (%)	n (%)	n (%)
Years’ experience in specialty				
Trainee	0 (0)	3 (18.8)	1 (11.1)	4 (10)
0–4	3 (20)	0 (0)	3 (33.3)	6 (15)
5–9	1 (6.7)	4 (26)	1 (11.1)	6 (15)
10–14	4 (26.7)	2 (12)	1 (11.1)	7 (18)
15–20	1 (6.7)	5 (31.3)	0 (0)	6 (15)
20+	6 (40)	2 (12)	3 (33.3)	11 (27)
Number of standard genomic tests previously ordered				
0	3 (20)	4 (26)	0 (0)	7 (17.5)
1–4	2 (13.3)	2 (12)	0 (0)	3 (7.5)
5–9	4 (26.7)	5 (31.3)	1 (11.1)	10 (25)
10–14	2 (13.3)	0 (0)	1 (11.1)	4 (10)
15–20	2 (13.3)	2 (12)	3 (33.3)	7 (17.5)
20+	2 (13.3)	3 (18.8)	4 (44.4)	9 (22.5)
Number of rapid genomic tests previously ordered				
0	5 (33.3)	6 (37.5)	0 (0)	11 (27.5)
1–4	6 (40)	7 (43.8)	2 (22.2)	15 (37.5)
5–9	2 (13.3)	2 (12)	2 (22.2)	6 (15)
10–14	1 (6.7)	0 (0)	3 (33.3)	4 (10)
15–20	0 (0)	0 (0)	0 (0)	0 (0)
20+	1 (6.7)	1 (6.3)	2 (22.2)	4 (10)
TOTAL	15	16	9	40

**Table 2 children-10-00970-t002:** Summary of intensivists’ responses of when explicit consent should be obtained for rGT (scenario A).

rGT Should/Should Not Proceed under the Following Circumstances:	Agree n (%)	Disagree n (%)
We should wait an extra day to allow the GHP obtain consent from family	20 (50)	20 (50)
We should wait for at least one parent’s consent before ordering rGT	30 (75)	10 (25)
rGT should NOT proceed if one parent has refused to provide consent	23 (57)	17 (43)
rGT should proceed if parents are overwhelmed but have given (possibly uninformed) consent	22 (55)	18 (45)
If the parents do not speak English rGT should NOT proceed until an interpreter is available	29 (72)	11 (28)

**Table 3 children-10-00970-t003:** Chi-squared tests of independence comparing responses of respondents by continent and by years of experience.

		Scenario A Questions					Scenario B Questions					
		1	2	3	4	5	1	2	3	4	5	6
Continent	Chi-squared	1.32	0.35	0.83	0.79	3.75	0.614	1.55	2.67	4.74	5.67	6.84
	DF	2	4	2	2	2	2	2	2	2	2	2
	*p*-value	0.518	0.987	0.66	0.675	0.153	0.736	0.461	0.263	0.093	0.059	0.033 *
Years Practising	Chi-squared	2.23	7.62	2.94	5.95	6.56	4.41	8.48	7.18	14.58	3.64	6.71
	DF	5	10	5	5	5	5	10	5	5	5	5
	*p*-value	0.816	0.666	0.71	0.311	0.256	0.492	0.582	0.207	0.012 *	0.601	0.243

* *p* < 0.05.

## Data Availability

Supporting data are available to bona fide researchers, subject to registration, from the UK Data Service.

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
