# Peer review of "Intensive Care Clinicians’ Perspectives on Ethical Challenges Raised by Rapid Genomic Testing in Critically Ill Infants"

_children, 2023, doi:10.3390/children10060970_

Round 1
Reviewer 1 Report
The manuscript is of great interest because it analyzes the attitude to urgent genomic research by specialists who make such decisions, namely neonatal/paediatric intensivists. This distinguishes this study from previously published works in which surveys of various specialists were conducted, but intensivists were not interviewed. The study of the opinion of this particular group of specialists is very important, since it is they who make the decision to provide assistance in these departments. The authors gathered the opinion of 40 experts from 12 different countries.
The authors did not reflect in the materials and methods section the three variants of diagnoses that they proposed in the case of scenario B. The fact that there are three options becomes clear only after reading the results. I propose to supplement the section with this information.
I did not have enough in the results and discussion of the results of the answers to scenario A. There is no discussion of these results.
The results obtained by the authors are new and interesting. It is very important information that the decision to use a genetic test and making clinical decisions based on its results depends on the experience of the doctor. The paper shows a serious and reliable influence of the results of genetic tests on the decision on the curation of patients and postulates the need to create guidelines on genetic tests for non-investigative intensivists.
I really liked this work, but I ask the authors to discuss in a more structured way all the results obtained according to the scenarios set in the study and supplement the materials and methods.
Author Response
Response to Reviewer 1 Comments
Point 1: The authors did not reflect in the materials and methods section the three variants of diagnoses that they proposed in the case of scenario B. The fact that there are three options becomes clear only after reading the results. I propose to supplement the section with this information.
Response to point 1: The methods section has been supplemented with information about the three types of diagnoses that are discussed in the survey.
Line 95-97: Questions about withholding treatment centred around three diagnoses (STRA-6 related disorders; Alagille syndrome and Kabuki syndrome) ranging in spectrum from mild to severe physical and intellectual impact.
Point 2: I did not have enough in the results and discussion of the results of the answers to scenario A. There is no discussion of these results.
Response to point 2: Section 3.2 is dedicated to the results of scenario A as well as section 4.1 of the discussion. Though these sections are slightly more concise than the sections about scenario B (3.3 and 4.2/4.3), this is due to prioritising the nuanced findings from responses to scenario B.
Point 3: I really liked this work, but I ask the authors to discuss in a more structured way all the results obtained according to the scenarios set in the study and supplement the materials and methods.
Response to point 3: The sub-headings in the results have been amended to explicitly state where responses from scenario A and B are being discussed. This, and the addition in the methods sections about the types of diagnoses, have been changed to outline the results and discussion of each scenario in a more structured way.
Line 156-157: 3.2. Obtaining parental consent in the context of Rapid Genomic Testing – responses to Scenario A
Line 218-219: 3.3. Withdrawal of life-sustaining treatment in the context of Rapid Genomic Testing – responses to Scenario B
Reviewer 2 Report
This study is a good effort to create consciousness and encourage future discussions about the ethical challenges that the implementation of rGT in a daily NICU/PICU setting generates.
The clinical scenarios in the survey could have included the very frequent settings where the information obtained from rGT creates uncertainty about the future, due to a lack of knowledge about the detected disease or even by detection of secondary findings. This would have enriched the study by assessing whether the intensivists ordering these tests acknowledge the real complexity of possible results that could be obtained and how they think this will impact their decision of ordering rGT regardless of parents' pre-test counseling and consent.
Noticed that on line 607: "Upon receiving the diagnosis of Kabuki syndrome, Sam’s parents decide they do not want to proceed with liver transplantation in childhood ...": Was this error present in the actual survey sent to participants? In that case, did the participants detect this mistake? This needs to be clarified.
Author Response
Response to Reviewer 2 comments:
Point 1: The clinical scenarios in the survey could have included the very frequent settings where the information obtained from rGT creates uncertainty about the future, due to a lack of knowledge about the detected disease or even by detection of secondary findings. This would have enriched the study by assessing whether the intensivists ordering these tests acknowledge the real complexity of possible results that could be obtained and how they think this will impact their decision of ordering rGT regardless of parents' pre-test counselling and consent.
Response to point 1: This is an interesting point. The complexity of genomic test results and the uncertainty they present is a very real issue in genomic testing. To increase the number of likely responses from intensive care clinicians, we made a decision to keep the survey very short. Unfortunately, this meant not exploring all elements related to prognostic uncertainty. We have now made this clear in the limitations section.
Line 384-386: This study did not include other challenging aspects of delivering rGT, such as how to counsel families around uncertain results.
Point 2: Noticed that on line 607: "Upon receiving the diagnosis of Kabuki syndrome, Sam’s parents decide they do not want to proceed with liver transplantation in childhood ...": Was this error present in the actual survey sent to participants? In that case, did the participants detect this mistake? This needs to be clarified.
Response to point 2: Thank you for picking up on this error. This was an error in transcribing the survey questions to the appendix. The appendix has been corrected to reflect the wording in the survey disseminated to participants.
Line 643-645: Upon receiving the diagnosis of Kabuki syndrome, Sam's parents decide they do not want to proceed with cardiac surgery and want their child to be transferred to palliative care. In your view, should parents have the authority to refuse treatment for their child in this case?
Reviewer 3 Report
The authors describe in this manuscript about a survey that was given to intensive care specialists about rapid whole genome sequencing.
I don't have any major concerns regarding this study.
Minor concerns:
What society guidelines (eg, ACMG, CCMG, ESHG, others) are available? It might be good to provide some of that information in the background/introduction. How does that impact this study?
My understanding is that geneticists are on call for most major children's hospitals when these children may be treated. (lines 46-56) While a GHP is not present in a 24/7 emergency setting, they should be accessible especially with telemedicine options. Perhaps discuss as part of introduction or conclusions.
I am not sure intensivists is a widely adopted term. Perhaps revise the title to intensive care specialists.
Line 59 - please confirm city. I believe the name should be Kansas City.
Author Response
Response to Reviewer 3 comments
Point 1: What society guidelines (eg, ACMG, CCMG, ESHG, others) are available? It might be good to provide some of that information in the background/introduction. How does that impact this study?
Response to point 1: We agree that discussion of what existing guidelines encompass and the gap that exists when discussing rGT provides important context for this discussion. Therefore we have added the following:
Line 74-81: Globally, there are several professional bodies that provide guidelines for genetic testing such as the American College of Medical Genetics and Genomics (AMCG), the Human Genetics Society of Australasia (HGSA), Canadian College of Medical Geneticists (CCMG) and the European Society of Human Genetics (ESHG). Each provide guidance for the use of genomic sequencing technologies in their region and establish ethical standards for practitioners when ordering predictive and diagnostic genetic testing. However, these bodies do not yet provide guidance for rGT and if recommendations for obtaining consent and storing DNA should differ in the rapid space [24].
Point 2: My understanding is that geneticists are on call for most major children's hospitals when these children may be treated (lines 46-56). While a GHP is not present in a 24/7 emergency setting, they should be accessible especially with telemedicine options. Perhaps discuss as part of introduction or conclusions.
Response to point 2: Lines 52-58 have been amended to include the factor of an on-call geneticist being available for consultations in many tertiary children’s hospitals. The language has been changed to emphasize that it will likely still be paediatric and neonatal intensivists that are required to obtain consent from families as rapid genomic testing becomes widely available.
Line 52-58: Currently, in many major children’s hospitals an on-call clinical geneticist is likely accessible for consultation via telehealth or telephone. However as rGT becomes more widely offered, paediatric and neonatal intensive care clinicians (who we refer to collectively as intensivists) will be the clinicians obtaining parental consent for rGT and make treatment decisions based on rGT outcomes without immediate support from GHPs. Furthermore, exact processes and professional responsibilities vary across hospitals and jurisdictions.
Point 3: I am not sure intensivists is a widely adopted term. Perhaps revise the title to intensive care specialists.
Response to point 3: The title has been amended to ‘Intensive Care Clinicians’ and the introduction amended to introduce this term.
Line 54-55: paediatric and neonatal intensive care clinicians (who we refer to collectively as intensivists)
Point 4: Line 59 - please confirm city. I believe the name should be Kansas City.
Response to point 4: Line 63 (formerly line 59) has been amended to read Kansas City.
Line 63: …experience using genomic technologies in Kansas City, Missouri, towards the use of rGT…